# Married women's decision-making autonomy in the household and maternal and neonatal healthcare utilization and associated factors in Debretabor, northwest Ethiopia

Azmeraw Ambachew Kebede[1]*, Endeshaw Admasu Cherkos[2], Eden Bishaw Taye[1], Getachew Azeze Eriku[3], Birhan Tsegaw Taye[4], Wagaye Fentahun Chanie[5]

1 Department of Clinical Midwifery, School of Midwifery, College of Medicine and Health Sciences, University of Gondar, Gondar, Ethiopia, 2 Department of Women's and Family Health, School of Midwifery, College of Medicine and Health Sciences, University of Gondar, Gondar, Ethiopia, 3 Department of Physiotherapy, School of Medicine, College of Medicine and Health Sciences, University of Gondar, Gondar, Ethiopia, 4 Department of Midwifery, College of Medicine and Health Sciences, Debre Berhan University, Debre Berhan, Ethiopia, 5 UNFPA Supported Maternal Health Project Coordinator, College of Medicine and Health Sciences, University of Gondar, Gondar, Ethiopia

* azmuzwagholic@gmail.com

**Data Availability Statement:** The datasets collected and analyzed for the current study are

## Abstract

### Background

Women's decision-making autonomy is very crucial for the improvement of women empowerment, and maternal, neonatal, and child healthcare utilization. As time immemorial, Ethiopian culture is largely gender stratified, and the position of women is subordinate to men in various household and health-seeking decision-making matters. However, there is a dearth of empirical evidence on women's decision-making autonomy, especially in the household and on maternal and newborn healthcare utilization. Therefore, this study assessed married women's decision-making autonomy in the household and on maternal and neonatal healthcare utilization and associated factors in Debretabor, Northwest Ethiopia.

### Methods

A community-based cross-sectional study was conducted from October 1st to 30th, 2019. A two-stage sampling technique was used to select 730 married women. Data were collected using a structured, pretested, and interviewer-administered questionnaire. Data were entered into EPI INFO 7 and analyzed by SPSS version 23. Multivariable logistic regression model was fitted to identify factors associated with women's decision-making autonomy in the household and maternal and neonatal healthcare utilization. The adjusted odds ratio with its 95% confidence interval was computed to determine the level of significance at a p-value of $\leq 0.05$.

### Results

A total of 730 married women were included in the analysis. Thus, three-fourths (75.1%) of women had higher decision-making autonomy on their health, neonatal health, and other

available within the manuscript and it's supporting information files.

**Funding:** We didn't gain any specific fund for this study.

**Competing interests:** We have no conflicts of interest to disclose.

**Abbreviations:** AOR, Adjusted Odds Ratio; ANC, Antenatal Care; CI, Confidence Interval; COR, Crude Odds Ratio; EDHS, Ethiopian Demographic Health Survey; ETB, Ethiopian Birr; MNCH, Maternal, Neonatal, and Child Health; NDS, Neonatal Danger Signs; PNC, Postnatal Care; SDG, Sustainable Development Goal; SPSS, Statistical Package for Social Science.

socio-economic aspects. Besides, the proportion of four and above antenatal visits, delivery at a health facility, postnatal checkup, knowledge of neonatal danger signs, and appropriate health-seeking practices for sick newborns among autonomous women were 52.1%, 56.1%, 71.4%, 32%, and 80% respectively. Age greater than 35 years old (AOR = 2.08; 95% CI: 1.19, 3.62), monthly income of > 5000 ETB (AOR = 3.1; 95% CI: 1.36, 7.07), husband involvement (AOR = 2.36; 95% CI: 1.55, 3.43), and adequate knowledge of neonatal danger signs (AOR = 2.11; 95% CI: 1.4, 3.2) were factors independently associated with women's decision-making autonomy on maternal and neonatal healthcare utilization and other socio-economic affairs.

## Conclusion

Our findings show that women's decision-making autonomy in the household and maternal and neonatal healthcare utilization was optimal. Increasing household income level, promotion of husband's involvement starting from the prenatal period, and increasing women's knowledge of maternal and neonatal danger signs will have a great role in the improvement of women's decision-making autonomy.

## Introduction

Women's autonomy is defined as the ability of women to act independently on their particular health, children's health, freedom of movement, and control over finance without requesting permission from somebody [1]. Empowering women is very essential for any socio-economic development of a country [2]. Hence, women's autonomy undoubtedly contributes to many health advantages for both the mother and their children. However, there is a dearth of a strong description of the concept and obtaining data on the individual, household, and community level that shows all opportunities of women empowerment [3]. Maternal and neonatal health provision needs a multi-sector approach that, in turn, requires a strong decision-making autonomy of women to reverse back the barriers at the household level [4].

Although maternal mortality has dropped by 2.9% every year between 2000–2017, there are still an estimated 295,000 maternal deaths and 2.6 million neonatal deaths in the year 2017 [5,6]. The strong decision-making power of women is vital to dropping this higher magnitude. Because, limited women's decision-making autonomy impedes maternal healthcare utilization such as antenatal care (ANC), postnatal care (PNC), and delivery at a facility [7]. Besides, the lower autonomy of women affects the socio-economic, emotional, fertility decision, contraceptive use, and the sexual life of the women [8,9]. Notably, decisions made at the household level affect not only the welfare of the individual but also the surrounding community even to the country level [10].

Studies have shown that the decision-making autonomy of women is low, specifically in developing countries. However, scaling up the women's role in a decision leads to better uptake of healthcare access, poverty reduction, and household economic growth [1,10,11]. In developing countries, women play an important role in the beneficence of the family, but they are essentially have seen as ordinary homemakers [12]. Literature in Ethiopia shows that maternal and neonatal health coverage is low [13,14]. Recent data revealed that 43%, 48%, and 34% of women had four and above ANC visits, give birth at a health facility, and postnatal checkup respectively [15]. According to the 2016 Ethiopian demographic health survey

(EDHS 2016), only 11–18% of women were involved in making decisions alone, and 66–68% together with their husband or partner [16]. Also, the neonatal mortality rate and the maternal mortality ratio of Ethiopia are 30 per 1000 live births [15] and 412 per 100,000 live births respectively [16].

So far, studies elsewhere in the world revealed that women's decision-making autonomy was 38.9% in Nigeria-[11], and parts of Ethiopia including Bale zone-41.4% [1], Ambo town-55.6% [17], Southern Ethiopia-58.4% [14], and analysis of 2011 EDH-54% [18] data revealed that women's decision-making autonomy was low; which was, respectively. Besides, studies from Iran, Nigeria, and elsewhere in Ethiopia including Ambo town, Southern Ethiopia, and Bale zone found that older maternal age, exposure to mass media, higher socioeconomic status, higher educational status, higher family size, and knowledge of maternal and child health were positively associated with women's decision-making autonomy [1,8,11,14,17].

Improving maternal and neonatal health is one of the government concerns both nationally and globally that comprises the third component of sustainable development goal (SDG) [19]. Improving the optimal health of both the mother and her neonate and typically decreasing the maternal and neonatal mortality as low as possible by assessing factors affecting maternal and neonatal healthcare utilization, specifically on the autonomy of women will undoubtedly have a significant role in the achievement of the 3rd SDG. Even though a few kinds of researches were conducted on the decision-making autonomy of women, some of the published studies failed to address some variables like husband involvement in maternal, neonatal, and child health (MNCH) services and knowledge of the maternal and neonatal illness. Besides, many of the previous studies focus on the effect of decision-making autonomy on maternal and neonatal healthcare utilization giving little attention to what factors deter women's decision-making autonomy in the household and on maternal and neonatal healthcare utilization.

In Ethiopia, there is a significant burden of maternal and neonatal death due to low healthcare utilization [15,16]. Promoting the decision-making autonomy of women is the mainstay to increase maternal and neonatal healthcare utilization. Therefore, this study assessed married women's decision-making autonomy in the household and on maternal and neonatal healthcare utilization and associated factors in Debretabor, Northwest Ethiopia.

## Methods

### Study design, setting, and period

A community-based cross-sectional study was conducted from October 1st to 30th, 2019 in Debretabor town, Northwest Ethiopia. The town is located 665 km Northwest of Addis Ababa (the capital city of Ethiopia) and 103 km Northeast of Bahir Dar (the capital city of Amhara regional state) and it is the capital city of South Gondar Zone. Currently, the town has a total population of 63,563, of whom 31,863 (54.8%) are female. An estimated 19,327 are women of the reproductive-age group. Of these, 13,936 women are currently in marital union. About 4317 (6.8%) are under five. Moreover, the town has one general hospital, three health centers, 6 health posts, and 6 private clinics serving the community (Debretabor administrative town report, unpublished data).

### Study population

All married women who had an infant age of one year and below and residing for at least six months in the selected 'kebeles were included in the study. Women who were seriously ill or mentally ill throughout the data collection period were excluded.

## Sample size determination & sampling procedures

The sample size for this study was determined by using a single population proportion formula by considering the following assumptions:- the proportion of women's decision-making autonomy on maternal and neonatal healthcare utilization 58.4% [14], 95% level of confidence, and 5% margin of error.

$$n = \frac{(Z\alpha/2)^2 p(1-p)}{(d)^2} = \frac{(1.96)^2 * 0.584(1-0.584)}{(0.05)2} = 374$$

Where n = required sample sizes, $\alpha$ = level of significance, z = standard normal distribution curve value for 95% confidence level = 1.96, p = proportion of women's decision-making autonomy on maternal and child healthcare utilization, d = margin of error. Finally, by considering a 10% non-response rate to minimize errors arising from the chance of non-compliance and a design effect of 2 (since 2 stage sampling was employed), the final calculated sample size was 748. In the first stage, three 'kebeles' were selected randomly by lottery method among the six 'kebeles', which is the smallest administrative unit in Ethiopia. We did a survey to identify all eligible women with the help of health extension workers (HEWs) and the sampling frame was designed by numbering the list of women using the registration book (we registered the house number of all eligible women). The calculated sample size was proportionally allocated to draw the study participants from each 'kebeles. Finally, the study participants were selected by a simple random sampling technique using a table of random generation.

## Variables of the study

The dependent variable for this study was women's decision-making autonomy whereas the explanatory variables were women's age, religion, women's educational level, women's occupation, husband occupation, husband educational level, husband involvement in MNCH services, family size, average monthly income (in Ethiopia Birr), media exposure, parity, ANC, number of ANC, place of delivery, assistant for the delivery, PNC, number of PNC, visited by health by HEWs (if the woman was visited by HEWs during pregnancy or after delivery at least once at home), history of neonatal death, time taken to the health facility (travel to the health facility on foot), and maternal knowledge of NDS.

## Measurement and operational definitions

**Women's decision-making autonomy.** For this study, it was composited to higher decision-making autonomy (which was coded as "1") and lower decision-making autonomy (which was coded as "0"). The women's responses were coded into (2 = for women who were able to decide individually, 1 = for women who were able to decide together with her husband, and 0 = otherwise). The cumulative score was ranged from 0 to 27 which is the minimum and maximum score respectively. Accordingly, women who scored above the mean were considered highly autonomous whereas women who scored below the mean were less autonomous [1,2].

The women have questioned "who in your household decides (1) Healthcare for yourself (2), Healthcare for newborns and/or children (3) Visit of family or relatives (4) To have additional children (5) Utilization of maternal and child healthcare services like ANC, PNC, and immunization (6) Large household purchases and consumptions (7) Intra-household resource allocation and purchases (8) Husband's earning and (9) Cooking daily foods. The likely replies for each question were women alone, together with her husband, or husband only.

**Good knowledge of NDS.**  Women who mentioned at least three neonatal danger signs among 12 neonatal danger signs [20].

**Appropriate health-seeking practices.**  Women who sought care for neonatal danger signs from well-qualified health professionals in governmental and/or private health facilities [21].

**Adequate ANC.**  Women who had four or more ANC visits in the most recent pregnancy [22].

**Facility delivery.**  Delivery in public or private hospitals and/or clinics in the most recent delivery [22].

**Had PNC.**  Women who received at least one postnatal checkup in the most recent delivery [22].

**Husband involvement in MNCH services.**  Based on the summative score of variables designed to assess husband involvement with a score above the mean was considered as involved [23].

**Exposure to media.**  Those women who responded at least once a week is considered to be regularly exposed to that form of media (television, radio, or magazine) [16].

## Data collection tools and procedures

The data collection tool was developed by reviewing related literature [1,2,18,24]. A structured, interviewer-administered questionnaire was employed to collect the data through face-to-face interviews. The content validity of the questionnaire was assessed by a group of senior researchers. Socio-demographic characteristics, reproductive and maternal health service characteristics, knowledge and healthcare-seeking practice, husband involvement, and decision-making autonomy-related characteristics were incorporated in the study tool. Three Diploma and Bachelor of Science in midwifery holders were recruited for data collection and supervision respectively.

## Data quality control

The questionnaire was first prepared in English and translated to the local language (Amharic) and back to English to keep its consistency and readjustments of inconsistent and inaccurate data were done accordingly. Before the actual data collection, a pretest was done on 5% of the sample size at Adis Zemen town, Northwest Ethiopia to check the response, language clarity, understanding of data collectors, and supervisors. During the actual data collection period, the questionnaire was checked for completeness daily by the supervisors and the principal investigator.

## Data processing and analysis

The collected data were checked for completeness manually and 18 participants were excluded from the analysis because of their incomplete data. Then, the data were checked, coded, and entered into Epidemiological Information (EPI INFO) version 7, and exported to Statistical Package for Social Sciences (SPSS) version 23 for analysis. Descriptive statistics like percentages, frequency, mean, standard deviation, tables, and graphs were used to present the characteristics of study participants. Pearson's chi-squared test was done to examine the association between individual-level factors and the outcome variable. The binary logistic regression model was fitted to identify risk factors for women's decision-making autonomy. Initially, bivariable analysis was performed to identify the candidate explanatory variables for the multivariable analysis. Thereafter, all explanatory variables having a p-value of $\leq 0.2$ in the bivariable analysis were included in the multivariable logistic regression analysis to handle the effect

of possible confounders and to identify independent factors affecting women's decision-making autonomy. Model fitness for the final model was checked using Hosmer and Lemeshow goodness of fit. Then, the level of significance was declared based on the adjusted odds ratio (AOR) with a 95% CI at a p-value of $\leq 0.05$.

### Ethical considerations

The study was done following the Ethiopian Health Research Ethics Guideline and the declaration of Helsinki. Ethical clearance was obtained from the school of Midwifery ethical review committee under the delegation of the University of Gondar Institutional Review Board (IRB). A formal letter of administrative approval was obtained from the Debretabor town health office. Also, anonymous written informed consent was taken from each of the study participants after a clear explanation of the aim of the study.

## Result

### Socio-demographic characteristics of study participants

A total of 730 married women were included in this study. Eighteen women were excluded from the study due to their incomplete data, giving a 98% response rate. The mean age of the participants was 30 years (SD ±5.86). Most of the study participants (97.9%) belonged to Orthodox Christian by religion. Almost half (49.2%) of the participants had accomplished a Diploma and above education. Half (50.8%) of women were housewives by occupation. Regarding the husband's occupation, 406 (55.6%) were government employees and two-thirds (66.7%) of them had completed college and above education [**Table 1**].

### Reproductive history and maternity healthcare service-related characteristics

From the total study participants, more than two-thirds (68.2%) of women had a parity of two to four. The majority of women (97.5%) had at least one ANC visit in their recent pregnancy, of whom, only 59.6% of women completed four ANC visits. Four-fifths (80.3%) of women gave birth at governmental health institutions for their recent delivery. Most (94%) of women had at least one postnatal visit for their most recent delivery [**Table 2**].

### Women's decision-making autonomy in the household and on maternal and neonatal healthcare utilization

The overall decision-making autonomy of women was found to be 75.1% (95% CI: 72.1, 78.1). About 72.1% of women had the joint decision with their husbands to visit the health facilities for their health when they become sick. More than four-fifths (84.8%) of the study participants decide with their husbands to take sick newborns to the health facility. Slightly, more than three-fourths (77.1%) and almost three-fourths (74.7%) of the participants had a joint decision with their husband for large household purchases and small intrahousehold resource allocation respectively [**Table 3**].

### Maternal and neonatal healthcare service utilization among autonomous women

The proportion of at least four ANC visits, delivery at a health facility, and appropriate health-seeking practices for sick neonates among autonomous women were 52.1%, 56.1%, and 80% respectively [**Fig 1**].

**Table 1. Socio-demographic characteristics of study participant in Debretabor, Northwest Ethiopia, 2019 (n = 730).**

| Charactestics | Frequency | Percentage (%) |
|---|---|---|
| **Age of women in year** | | |
| 18–24 | 112 | 15.3 |
| 25–34 | 418 | 57.3 |
| ≥ 35 | 200 | 27.4 |
| **Religion** | | |
| Orthodox Christian | 708 | 96.99 |
| Muslim | 13 | 1.78 |
| Protestant | 9 | 1.23 |
| **Educational status of the women** | | |
| No formal education | 80 | 11 |
| Primary education | 131 | 17.9 |
| Secondary education | 160 | 21.9 |
| Diploma and above | 359 | 49.2 |
| **Occupation of the women** | | |
| House wife | 371 | 50.8 |
| Government employee | 217 | 29.7 |
| Self employed | 26 | 3.6 |
| Merchant | 87 | 11.9 |
| Others [a] | 29 | 4 |
| **Husband educational status** | | |
| No formal education | 24 | 3.3 |
| Primary | 63 | 8.6 |
| Secondary | 156 | 21.4 |
| Diploma and above | 487 | 66.7 |
| **Husband occupation** | | |
| Government employee | 406 | 55.62 |
| Merchant | 155 | 21.23 |
| Self employed | 115 | 15.75 |
| Daily labor | 45 | 6.16 |
| Others [b] | 9 | 1.24 |
| **Exposure to media** | | |
| Yes | 639 | 87.5 |
| No | 91 | 12.5 |
| **Family size** | | |
| <3 | 155 | 21.2 |
| 3–5 | 533 | 73 |
| >5 | 43 | 5.8 |
| **Average monthly income of the family** | | |
| ≤ 1200 ETB | 33 | 4.5 |
| 1201–3000 ETB | 242 | 33.2 |
| 3001–5000 ETB | 246 | 33.7 |
| >5000 ETB | 209 | 28.6 |

a = student and daily labour

b = student and carpenter.

**Table 2. Reproductive history and maternity healthcare service-related characteristics of study participant in Debretabor, Northwest Ethiopia, 2019 (n = 730).**

| Characteristics | Frequency | Percentage (%) |
|---|---|---|
| **Parity** | | |
| 1 | 197 | 27 |
| 2–4 | 483 | 68.2 |
| >4 | 50 | 6.8 |
| **Had ANC** | | |
| Yes | 712 | 97.5 |
| No | 18 | 2.5 |
| **Number of ANC follow up (n = 712)** | | |
| <4 | 288 | 40.4 |
| ≥4 | 424 | 59.6 |
| **Place of delivery** | | |
| Government hospital | 586 | 80.3 |
| Health center | 99 | 13.6 |
| Private hospital/clinic | 11 | 1.5 |
| At home | 34 | 4.7 |
| **Birth assistant** | | |
| Health professionals | 697 | 95.5 |
| TBAs | 33 | 4.5 |
| **Had PNC** | | |
| Yes | 686 | 94 |
| No | 44 | 6 |
| **Number of PNC (n = 686)** | | |
| <3 | 279 | 40.7 |
| ≥3 | 407 | 59.3 |
| **Husband involvement** | | |
| Involved | 338 | 46.3 |
| Not involved | 392 | 53.7 |
| **Visited by HEWs** | | |
| Yes | 388 | 53.2 |
| No | 342 | 46.8 |
| **History of neonatal death** | | |
| Yes | 25 | 3.4 |
| No | 705 | 96.6 |
| **Time taken to the health facility** | | |
| <30 minute | 717 | 98.2 |
| ≥ 30 minute | 13 | 1.8 |

ANC = antenatal care, HEWs = health extension workers, PNC = postnatal care.

## Factors associated with women's decision-making autonomy in the household and on maternal and neonatal healthcare utilization

From the multivariable logistic regression analysis age greater than 35 years old, higher monthly income of the family (> 5000 ETB), adequate knowledge of NDS, and husband's involvement (those involved) in MNCH related activities had a significant association with women's decision-making autonomy.

**Table 3. Women's household decision-making autonomy characteristics in Debretabor, northwest Ethiopia, 2019.**

| Variables | Frequency | Percentage (%) |
|---|---|---|
| **Decision on her own health** | | |
| Self | 61 | 8.4 |
| Jointly | 526 | 72.1 |
| Husband | 143 | 19.6 |
| **Decision on large household purchase** | | |
| Self | 15 | 2.1 |
| Jointly | 563 | 77.1 |
| Husband | 152 | 20.8 |
| **Decision on small household purchase** | | |
| Self | 156 | 21.4 |
| Jointly | 545 | 74.7 |
| Husband | 29 | 4 |
| **Decision on children and/or neonatal health** | | |
| Self | 37 | 5.1 |
| Jointly | 619 | 84.8 |
| Husband | 74 | 10.1 |
| **Decision on to have additional child** | | |
| Self | 27 | 3.7 |
| Jointly | 625 | 85.6 |
| Husband | 78 | 10.7 |
| **Decision on visiting to family and/or relatives** | | |
| Self | 113 | 15.5 |
| Jointly | 546 | 74.8 |
| Husband | 71 | 9.7 |
| **Decision on husband's earning** | | |
| Self | 15 | 2.1 |
| Jointly | 496 | 67.9 |
| Husband | 219 | 30 |
| **Decision on cooking daily foods** | | |
| Self | 498 | 68.2 |
| Jointly | 202 | 27.7 |
| Husband | 30 | 4.1 |
| **Decision to utilize maternal health services** | | |
| Self | 77 | 10.55 |
| Jointly | 620 | 84.93 |
| Husband | 33 | 4.52 |

Those women greater than 35 years old were two times (AOR = 2.08; 95% CI: 1.19, 3.62) more likely to have had higher decision-making autonomy compared to the reference group (18–24 years old). The odds of having higher decision-making autonomy among women who had a monthly income of > 5000 ETB was three times higher as compared to women who had a monthly income of ≤ 1200 ETB (AOR = 3.10; 95% CI: 1.36, 7.07). Similarly, those women who got husband involvement in MNCH related activities were 2.36 times (AOR = 2.36; 95% CI: 1.60, 3.47) more likely to be autonomous as compared to those women who didn't have husband support. This study also revealed that those women who had adequate knowledge of neonatal danger signs were two times (AOR = 2.11; 95% CI: 1.4, 3.2) more likely to have had

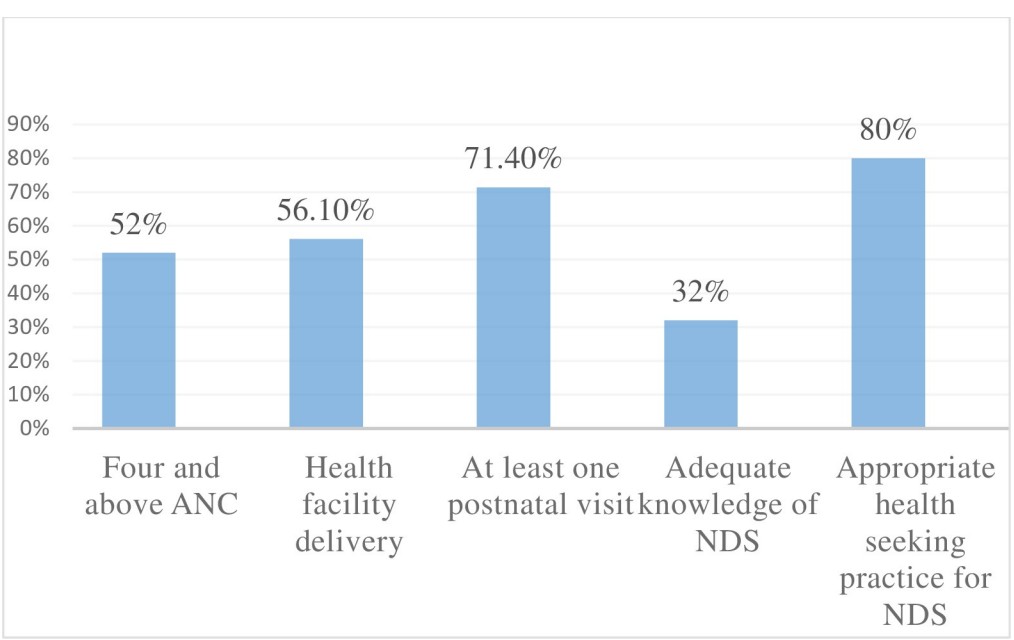

**Fig 1. Maternal and neonatal healthcare service utilization among autonomous women in Debretabor northwest Ethiopia, 2019.**

higher decision-making autonomy as compared to women who had poor knowledge of newborn danger signs [**Table 4**].

## Discussion

Women's decision-making autonomy is very important for the wellbeing of the family, particularly for the improvement of maternal and neonatal health in resource-limited countries like Ethiopia. This study assessed decision-making autonomy in the household and on maternal and neonatal healthcare utilization among married women who had an infant of one year and below in Debretabor, Northwest Ethiopia. Accordingly, three-fourths of women had higher decision-making autonomy in the household and on maternal and neonatal healthcare utilizations. Furthermore, this study found that the proportion of four and above ANC visits, delivery at a health facility, at least one postnatal checkup, good knowledge of NDS, and appropriate health-seeking practice for sick newborns among autonomous women was 52.1%, 56.1%, 71.4%, 32%, and 80% respectively.

The current study revealed that the overall women's decision-making autonomy in the household and on maternal and neonatal healthcare utilization was 75.1% (95% CI: 72.1, 78.1). This finding is in line with a study conducted in Ghana-75% [25]. However, the finding of this study was higher than previous studies conducted in Nigeria-38.9% [11], Ghana-52.8% [26], and Senegal-6.26% [27]. This could be possibly explained by differences in socio-demographic characteristics and study population. The study population in this study were married women in which their decision-making autonomy is expected to be higher compared with their unmarried counterparts. Two-thirds of women from the Senegal study were unmarried that might be responsible for the lower proportion of women's decision-making autonomy. Besides, nearly two-fifths (37%) of women from Ghana and half (50.9%) of women from Senegal have no formal education. Furthermore, our study was conducted in a specific area, whereas the result from Nigeria was from the national demographic health survey. The result

**Table 4. Bivariable and multivariable logistic regression analysis of factors associated with women's decision-making autonomy in Debretabor, Northwest Ethiopia, 2019 (n = 730).**

| Variables | Decision-making autonomy | | COR (95%CI) | AOR (95%CI) |
|---|---|---|---|---|
| | Higher decision-making autonomy | Lower decision- making Autonomy | | |
| **Age of women** | | | | |
| 18–24 | 70 | 42 | 1 | 1 |
| 25–34 | 318 | 100 | 1.9 (1.22, 2.97) | 1.59(0.98, 2.56) |
| ≥ 35 | 160 | 40 | 2.4 (1.43, 4.02) | 2.08 (1.19, 3.62)* |
| **Women educational status** | | | | |
| No formal education | 56 | 24 | 1 | 1 |
| Primary education | 91 | 40 | 0.98(0.53, 1.78) | 0.99 (0.5, 1.89) |
| Secondary education | 111 | 49 | 0.97(0.54, 1.74) | 0.94 (0.5, 1.87) |
| Diploma and above | 290 | 69 | 1.8 (1.04, 3.11) | 0.85 (0.45, 1.64) |
| **Husband involvement** | | | | |
| Involved | 287 | 51 | 2.82 (1.96, 5.7) | 2.36 (1.55, 3.43) ** |
| Not involved | 261 | 131 | 1 | 1 |
| **Knowledge of NDS** | | | | |
| Knowledgeable | 234 | 41 | 2.56(1.74, 3.77) | 2.11(1.4, 3.2) ** |
| Not knowledgeable | 314 | 141 | 1 | 1 |
| **Media exposure** | | | | |
| Exposed | 493 | 146 | 2.21 (1.4, 3.5) | 0.99 (0.57, 1.74) |
| Not exposed | 55 | 36 | 1 | |
| **Parity** | | | | |
| 1 | 135 | 62 | 1 | 1 |
| 2–4 | 373 | 110 | 1.56(1.08, 2.25) | 1.23 (0.77, 1.95) |
| > 4 | 40 | 10 | 1.84(0.87, 3.91) | 1.13 (0.46, 2.77) |
| **Income** | | | | |
| ≤ 1200 ETB | 16 | 17 | 1 | |
| 1201–3000 ETB | 173 | 69 | 2.66(1,27, 5.57) | 2.68 (1.23, 5.57) * |
| 3001–5000 ETB | 187 | 59 | 3.37(1.6, 7.1) | 2.37(1.07, 5.22) * |
| >5000 ETB | 172 | 37 | 4.9 (2.3, 10.6) | 3.1(1.36, 7.07)* |

* P value <0.001

** p value <0.05, 1 reference category, ETB = Ethiopian Birr, NDS = Neonatal danger signs.

of the current study is also higher as compared to previous studies conducted somewhere else in Ethiopia including Bale zone-41.4% [1], Ambo town-55.6% [17], Wollaita and Dawro zones-58.4% [14], and analysis from EDHS 2011–54% [18]. This variation might be due to the differences in the time gap, and the study population's socio-demographic characteristics. More than fourth-fifths (86.7%) of participants from the Bale zone, 100% of participants from Wollaita and Dawro zones, and 81% of participants from the 2011 EDHS were from the rural area. However, the present study was conducted from the urban population, in which the habit of attending information and realizing it is higher in urban areas. Besides, the high proportion of women's decision-making autonomy in the current study might be due to the presence of enabling factors like educational status in which nearly half of the study participants complete college and above. Women's educational status is found to be an enabling factor for decision-making autonomy from studies elsewhere [28,29]. For instance, nearly one-third of women from the Bale zone, nearly two-fifths from Wollaita and Dawro zones, and slightly more than two-thirds of the EDHS data have no formal education that might be responsible for the variation.

The finding of this study, however, is lower as compared to a study conducted in Basoliben district, Ethiopia in which 80% of women had higher decision-making autonomy regarding family planning [30]. It is also lower than a finding from further analysis of the EDHS 2016 in which 81.6% of women had higher decision-making autonomy. The possible explanation for the differences might be due to differences in study participant's socio-economic characteristics, outcome variable measurement, and study population. In our study, nearly two-thirds of the study participants had a monthly income of > 3000 ETB whereas only 14.7% of women from Basoliben district have a monthly income of > 1500 ETB, and 38.5% of women from the EDHS 2016 data belonged to the poorest wealth status. In this study, it has been noted that higher household monthly income was significantly associated with women's decision-making autonomy. Moreover, the aforementioned studies use one component of reproductive health services (family planning for the Basoliben district and maternal healthcare utilization for the EDHS data) to measure decision-making autonomy whereas our study uses the different aspect of women's decision-making participations like freedom of movement, financial freedom, and decision-making ability for healthcare utilization.

The current study affirmed that maternal age greater than 35 years old were two times more likely to have had higher decision-making autonomy as compared to their counterparts. This finding is corroborated by other studies conducted in Zambia [31] and Southern Ethiopia [14], in which women greater than 30 years old were more likely to be participating in the decisions on their health and other household activities. This might be due to as age gets increased their educational status may have also increased and respect between couples will increase with age. Also, there is more conservative tradition and societal norms for women of the same age with their husbands in most developing countries in Ethiopia. Consequently, women get attention and recognition for household decision-making after they get older [32,33]. Likewise, adequate knowledge of NDS plays a great role in the decision-making autonomy of women in the household and on maternal and neonatal health. Thus, this study revealed that women who had adequate knowledge of neonatal danger signs were two times more likely to have had higher decision-making autonomy as compared to women who had poor knowledge of neonatal danger signs. This finding is consistent with a study conducted in Bale zone, Ethiopia, in which women who had adequate knowledge of maternal and child health were independently associated with women's autonomy on maternal and child healthcare utilization [1]. This may be because having good knowledge of maternal and neonatal health enforces women to challenge their husbands because they comprehend the seriousness of the illness.

Similarly, this study found that higher monthly income had a direct association with women's decision-making autonomy in the household and on maternal and neonatal healthcare utilization. Women who had a monthly income of > 5000 ETB were three times more likely to have had a higher decision-making autonomy for their health, neonatal health, and other soci0-economic activities as compared to women who had a monthly income of ≤ 1200 ETB. This finding is in agreement with a study conducted in Southern, Ethiopia [14]. It is also in accordance with a study done in Zambia, that founds women in the higher wealth index category were more likely to participate in the decision-making process regarding maternal healthcare utilization and household tasks [31]. This is possibly due to the fact that economically capable women are more likely to use mass media devices (i.e. television, radio, and magazine), and attend different meetings outside the home compared with the poor population. In turn, awareness regarding their rights and gender equality should be ensured thereby increasing, (maternal and neonatal health services utilization. Hence, policymakers and other stakeholders need to give special attention to increasing job opportunities for women, equal resource sharing with men, and low interest loan grants to build women's empowerment in all aspects.

Moreover, husband involvement in MNCH services has been strongly associated with women's decision-making autonomy in the household and maternal and neonatal healthcare utilization. Thus, women who got husband involvement in MNCH services were 2.36 times more likely to have had higher decision-making autonomy as compared to women who hadn't get husband involvement. Studies elsewhere reported that husband involvement encourages women's participation in socio-economic activities [34], associated with women's use of skilled maternal and neonatal health services [35], better intra-spousal communication, birth preparedness, and readiness for complications and utilization of maternal health services that, in turn, improves women's decision-making autonomy and a shared decision between couples [11,36]. The other explanation might be the certainty of the women will increase because of being helped and encouraged by their husbands and accordingly increased shared decision. This suggests that mass media agencies and non-governmental organizations better raise awareness regarding the right of women and the need for husband involvement in household activities, reproductive and sexual health, and maternal and neonatal healthcare access. Moreover, strategies should be designed to minimize male dominancy in and out of the home.

## Limitation of the study

We are pleased to acknowledge the limitations of this study. First, social desirability bias may not be eliminated since the study was self-reported. However, a better way of understanding was ensured by trained data collectors about the significance of the study and their participation can play an incredible effect on the benefits of the study findings. Second, it is impossible to infer cause-effect relationships between the outcome variable and the identified explanatory variables due to the cross-sectional nature of the study.

## Conclusion

Women's decision-making autonomy in the household and maternal and neonatal healthcare utilization was optimal. Being older age, having a higher monthly income, being knowledgeable on NDS, and getting husband involvement in MNCH were the factors that contribute women to have had a higher decision-making autonomy in the household and on maternal and neonatal healthcare utilization. Thus, great attention should be paid by the concerned body to modifiable factors like household income level, knowledge level of women, and husband's involvement in MNCH services starting from the prenatal period to increase women's decision-making autonomy. This, in turn, would be helpful to increase maternal and neonatal healthcare service utilization.

## Supporting information

**S1 File. Amharic and English version of the questionnaire.**
(DOCX)

**S2 File. SPSS dataset.**
(SAV)

## Acknowledgments

We would like to thank the University of Gondar for providing study ethical clearance. Our thankfulness also goes to all data collectors and study participants. We are glad to Debretabor Town Health Office for giving us a permission letter.

## Author Contributions

**Conceptualization:** Azmeraw Ambachew Kebede.

**Data curation:** Azmeraw Ambachew Kebede, Endeshaw Admasu Cherkos, Eden Bishaw Taye, Getachew Azeze Eriku, Birhan Tsegaw Taye, Wagaye Fentahun Chanie.

**Formal analysis:** Azmeraw Ambachew Kebede, Endeshaw Admasu Cherkos, Eden Bishaw Taye, Getachew Azeze Eriku, Birhan Tsegaw Taye, Wagaye Fentahun Chanie.

**Investigation:** Azmeraw Ambachew Kebede, Endeshaw Admasu Cherkos, Eden Bishaw Taye, Getachew Azeze Eriku, Birhan Tsegaw Taye, Wagaye Fentahun Chanie.

**Methodology:** Azmeraw Ambachew Kebede, Endeshaw Admasu Cherkos, Eden Bishaw Taye, Getachew Azeze Eriku, Birhan Tsegaw Taye, Wagaye Fentahun Chanie.

**Validation:** Azmeraw Ambachew Kebede, Endeshaw Admasu Cherkos, Eden Bishaw Taye, Getachew Azeze Eriku, Birhan Tsegaw Taye, Wagaye Fentahun Chanie.

**Visualization:** Azmeraw Ambachew Kebede, Endeshaw Admasu Cherkos, Eden Bishaw Taye, Getachew Azeze Eriku, Birhan Tsegaw Taye, Wagaye Fentahun Chanie.

**Writing – original draft:** Azmeraw Ambachew Kebede.

**Writing – review & editing:** Azmeraw Ambachew Kebede, Endeshaw Admasu Cherkos, Eden Bishaw Taye, Getachew Azeze Eriku, Birhan Tsegaw Taye, Wagaye Fentahun Chanie.

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
