## [Decision Letter · Decision Letter 0]

21 May 2021

PONE-D-20-25863

Married Women’s Decision-Making Autonomy on Maternal and Neonatal Healthcare Utilization and Associated Factors in Debretabor, Northwest Ethiopia

PLOS ONE

Dear Dr. Kebede,

Thank you for submitting your manuscript to PLOS ONE. After careful consideration, we feel that it has merit but does not fully meet PLOS ONE’s publication criteria as it currently stands. Therefore, we invite you to submit a revised version of the manuscript that addresses the points raised during the review process.

The manuscript has been evaluated by three reviewers, and their comments are available below.

The reviewers have raised a number of major concerns. They particularly note the need for clearer reporting of the results and request further rigor applied to the statistical analyses. In addition, they note that greater depth of discussion and organization is required with regards to the literature review.

We look forward to receiving your revised manuscript.

Kind regards,

Avanti Dey, PhD

Staff Editor

PLOS ONE

Journal Requirements:

2. Please include additional information regarding the survey or questionnaire used in the study and ensure that you have provided sufficient details that others could replicate the analyses. For instance, if you developed a questionnaire as part of this study and it is not under a copyright more restrictive than CC-BY, please include a copy, in both the original language and English, as Supporting Information.  If the original language is written in non-Latin characters, for example Amharic, Chinese, or Korean, please use a file format that ensures these characters are visible.

3. Please state whether you validated the questionnaire prior to testing on study participants. Please provide details regarding the validation group within the methods section.

[We would like to thank the University of Gondar for providing study ethical clearance and financial support.]

 [No funding. We are from a low income country, Ethiopia. ]

Reviewers' comments:

Reviewer's Responses to Questions

**Comments to the Author**

1. Is the manuscript technically sound, and do the data support the conclusions?

Reviewer #1: No

Reviewer #2: Yes

Reviewer #3: Partly

2. Has the statistical analysis been performed appropriately and rigorously? 

Reviewer #1: No

Reviewer #2: Yes

Reviewer #3: Yes

3. Have the authors made all data underlying the findings in their manuscript fully available?

Reviewer #1: Yes

Reviewer #2: Yes

Reviewer #3: Yes

4. Is the manuscript presented in an intelligible fashion and written in standard English?

Reviewer #1: No

Reviewer #2: Yes

Reviewer #3: Yes

5. Review Comments to the Author

Reviewer #1: 1. The literature review is inadequate and not properly organized.

2. The statistical analysis is not rigorous and no regression model is explicitly specified.

3. Results are not reported and explained clearly.

4. The manuscript is written in poor English

Reviewer #2: Overall, the paper provides important evidence regarding the decision-making process on the utilisation of maternal and neonatal health services. I do, however, have comments for some parts of the paper.

In many parts of the world, parents and other family members, especially mothers or grandmother play an important role in women's decision making on healthcare use. Authors should clarify why other family members were not included in the response of the decision-making variable.

A number of independent variables (e.g. time taken to health facility, visited by HEWs and others) which were reported in the table should also be described in the methods section.

It is important to present the proportion of healthcare practices and knowledge among autonomous women in table or figure.

Study found that women with adequate knowledge of neonatal dangers signs tend to have higher decision-making autonomy compared to those with poor knowledge. However, the proportion of autonomous women with good knowledge of neonatal danger signs was only 32%. This should be clarified.

Husband involvement is one factor associated with women decision-making autonomy. Should prenatal care also target husband to increase their awareness and support to mothers? It would be good to suggest more practical recommendations on how to improve the decision-making process within the household.

Reviewer #3: I have made the necessary comments on the copy of the manuscript which is attached. I did this because it will be difficult for me to refer the author around the paper without the line numbers on the manuscript

6. PLOS authors have the option to publish the peer review history of their article (what does this mean?). If published, this will include your full peer review and any attached files.

Reviewer #1: No

Reviewer #2: No

Reviewer #3: **Yes: **Phillips Obasohan

---

## [Decision Letter · Decision Letter 1]

9 Jul 2021

Married women’s decision-making autonomy in the household and maternal and neonatal healthcare utilization and associated factors in Debretabor, northwest Ethiopia

PONE-D-20-25863R1

Dear Dr. Kebede,

We’re pleased to inform you that your manuscript has been judged scientifically suitable for publication and will be formally accepted for publication once it meets all outstanding technical requirements.

Kind regards,

Frank T. Spradley

Academic Editor

PLOS ONE

Reviewers' comments:

Reviewer's Responses to Questions

**Comments to the Author**

1. If the authors have adequately addressed your comments raised in a previous round of review and you feel that this manuscript is now acceptable for publication, you may indicate that here to bypass the “Comments to the Author” section, enter your conflict of interest statement in the “Confidential to Editor” section, and submit your "Accept" recommendation.

Reviewer #1: (No Response)

Reviewer #2: All comments have been addressed

Reviewer #3: All comments have been addressed

2. Is the manuscript technically sound, and do the data support the conclusions?

Reviewer #1: (No Response)

Reviewer #2: Yes

Reviewer #3: Yes

3. Has the statistical analysis been performed appropriately and rigorously? 

Reviewer #1: (No Response)

Reviewer #2: Yes

Reviewer #3: Yes

4. Have the authors made all data underlying the findings in their manuscript fully available?

Reviewer #1: (No Response)

Reviewer #2: Yes

Reviewer #3: Yes

5. Is the manuscript presented in an intelligible fashion and written in standard English?

Reviewer #1: (No Response)

Reviewer #2: Yes

Reviewer #3: Yes

6. Review Comments to the Author

Reviewer #1: (No Response)

Reviewer #2: Thank you for your response to my queries and the clarifications provided. I have no further questions or comments.

Reviewer #3: The authors have addressed all the points raise in the review. Good work done at the revision and hope the recommendations provided will be taken into consideration by relevant bodies

7. PLOS authors have the option to publish the peer review history of their article (what does this mean?). If published, this will include your full peer review and any attached files.

Reviewer #1: No

Reviewer #2: No

Reviewer #3: **Yes: **Phillips Edomwonyi Obasohan

---

## [Editor Report · Acceptance letter]

15 Sep 2021

PONE-D-20-25863R1 

Married women’s decision-making autonomy in the household and maternal and neonatal healthcare utilization and associated factors in Debretabor, northwest Ethiopia 

Dear Dr. Kebede:

I'm pleased to inform you that your manuscript has been deemed suitable for publication in PLOS ONE. Congratulations! Your manuscript is now with our production department. 

Kind regards, 

on behalf of

Dr. Frank T. Spradley 

Academic Editor

PLOS ONE